# Methodologies for Determining the Service Quality of the Intercity Rail Service Based on Users' Perceptions and Expectations in Thailand

**Sajjakaj Jomnonkwao *** , **Thanapong Champahom** and **Vatanavongs Ratanavaraha**

School of Transportation Engineering, Institute of Engineering, Suranaree University of Technology, Nakhon Ratchasima 30000, Thailand; thanapongbas3004@gmail.com (T.C.); vatanavongs@g.sut.ac.th (V.R.)
* Correspondence: sajjakaj@g.sut.ac.th; Tel.: +66-4422-4251; Fax: +66-4422-4608

**Abstract:** There is a significant need to change people's travel mode from personal cars to public rail, because rail transport is a more environmentally friendly travel mode. Over the past decade, the number of rail passengers has reduced because of service quality problems. Thus, this study aims to propose guidelines for precise service quality (SQ) improvements of intercity rail services in Thailand. Data were collected from 615 train passengers by distributing questionnaires at train stations in six provinces, covering all regions of Thailand. Cluster analysis (CA), factor analysis (FA), and importance-performance analysis (IPA) were applied in this research, which were used based on gap analysis. As a result of CA and FA, the 45 quality indicators were grouped into four factors, namely, vehicles, staff, services, and infrastructures/stations. The FA results seem more appropriate than those of CA in terms of providing factor loadings that indicate the importance of each indicator. The results of IPA show that the seven indicators that were analyzed fell into the "concentrate here" quadrant. To summarize the current policy, the factor most in need of rapid improvement in order to increase the quality of the intercity rail service in Thailand is that of the train car variables group; on the other hand, the main strength of the current services relates to the services provided by staff.

**Keywords:** gap analysis; cluster analysis; factor analysis; importance-performance analysis (IPA); Thai railway; customer satisfaction survey

---

## 1. Introduction

### 1.1. The Rail Transportation Situation in Thailand

There are three modes of intercity transportation in Thailand, namely, bus, rail, and air transportation. The largest portion of intercity transportation is controlled by buses (85%), followed by rail (7%) and air (6%) transportation [1]. According to the Asian Development Bank [2], Thailand has, for the last 30 years, focused on the development of road transportation. The railway network is very small compared to the road network, as shown in Figure 1. However, when considering the objective of sustainable transportation, road transportation should be reduced because of the higher possibility of accidents and the higher transportation costs compared to other types of transportation [3]. Many developed countries, such as Japan and Germany, strongly support intercity rail system transportation as the major mode, because it reduces costs and greenhouse gases compared to other modes of transportation [2,4,5]. Previous studies have focused on the increasing number of public transport passengers. One of the widely utilized methods is the customer satisfaction survey, which is a survey of the current attitudes of passengers to the provided services [6]. There are various dimensions of the indicators, namely, vehicle, safety, information guidance, and service staff [7–9]. The output of such studies comprises policy or guidance to increase passenger satisfaction, and many

research works have concluded that word of mouth or customer loyalty increases, and the number of public transportation passengers increases in response to these interventions [10,11].

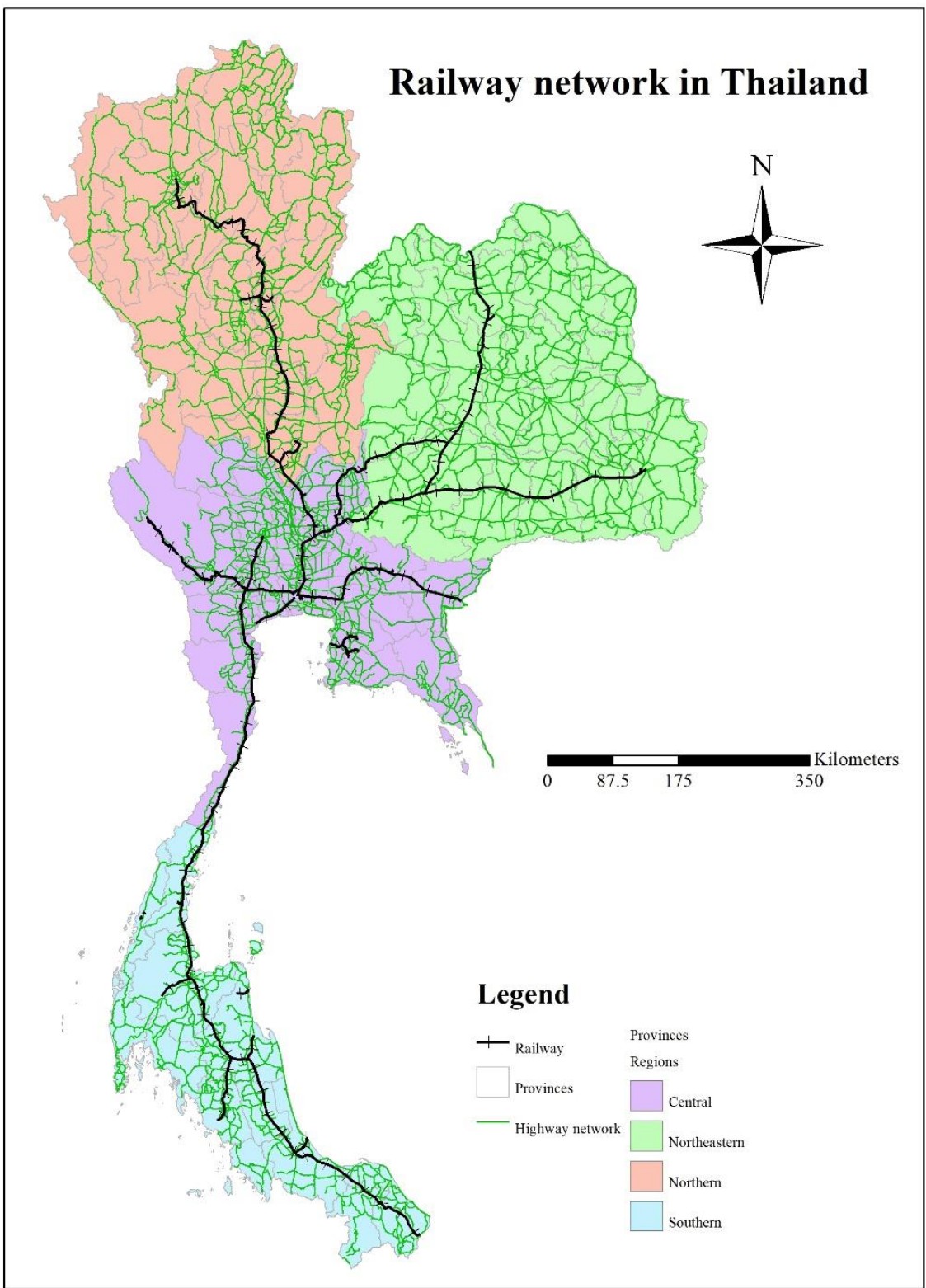

**Figure 1.** Railway network in Thailand. Note: Railway network excludes railways that are under construction.

## 1.2. Intercity Rail Service Quality Attributes

In the process of questionnaire creation, the concept of indicator development in terms of railway travel service quality can be classified simply into many categories, as follows:

Vehicles: these are indicators of both the interior and the exterior of the train car. The questions relate to a decent vehicle body appearance, a convenient interior temperature, clean toilets, clean train interior, space between seats, compartments, and availability of onboard food/drink [8].

Staff: this includes appropriate dressing attire, correct provision of services, ability to solve immediate onboard problems, and fast and accurate delivery of service [12].

Services: these include punctuality, ticket prices, number of ticket counters, and safety [12].

Information guidance: this includes traveling data, information delivery channels, and complaint channels.

Station: examples of indicators include cleanliness of the station, sufficient seating inside the station, adequate car parking facilities, and the availability of a security system and infrastructure such as Wi-Fi [13–15].

## 1.3. The Methodologies for Determining the Quality of Transportation Services

From customer survey data, many widely used analytical methods have been used to extract knowledge and to create policy recommendations for increasing passenger satisfaction. For example, first, in correlation analysis between qualitative factors toward satisfaction, the methods that have been used include the regression model, path analysis, and structural equation modeling (SEM). These methods have been used to find a linear relationship to search the most appropriate factor influencing satisfaction, after which policy has been developed [9,11,16]. Second, indicator grouping is related to theory for extracting weighted values. The methods that have been used include exploratory factor analysis and confirmatory factor analysis, which aim to find the correlation between indicators. This correlation can create a new latent factor, while the loading factor can be considered to select indicators, which can lead to policy definition [7,17–19]. Third, indicator group forming from the correlation and average of indicators without parameter estimation (as in examples 1 and 2 above) have been used. Examples of methods that have been used in this analytical group include cluster analysis (CA), the objective of which is to divide the observations into homogeneous and distinct groups [20], and importance–performance analysis (IPA) [21]. These methods have been used to plot the dimensions (importance and performance) of each indicator for grouping into four groups. The group with high importance and low performance, referred to as "concentrate here", is intended to undergo urgent improvement of efficiency in order to increase the operational performance of the public transportation system [22].

From the above methods, research has found that the methods in the first group are needed to survey the overall satisfaction of respondents, such as "overall level of satisfaction with transport service" [18]. However, as discussed by Ograjenšek and Gal [23], there is a weakness in this question in that respondents may understand the word "overall" from different perspectives. Thus, more specific questions can better reduce the bias of the questionnaire.

## 1.4. Research Gap and Objective of This Study

Based on the literature review, it was observed that all of the previous research works [9,11,16–22] applied only one or two analysis methods in terms of factor analysis (FA) and grouping of indicators. For example, the study of Ratanavaraha et al. [11] applied exploratory factor analysis (EFA) and utilized structural equation modeling (SEM). The study of Watthanaklang et al. [17] was similar, with the addition of confirmatory factor analysis (CFA). For studies using cluster analysis (CA), Cabral et al. [20] applied only one CA method. For an example of research work that applied only importance–performance analysis (IPA), Champahom et al. [22] used IPA alone to establish a quality

improvement roadmap for public bus services. From previous well-known analysis methods, no studies have focused on the comparison between the three above-mentioned methods.

Therefore, this study applies three methods, namely, CA, FA, and IPA, to find development guidelines by precise service quality improvement in order to increase the number of railway passengers. The analysis results are discussed in terms of operational function, which future research can use as a study guideline. The contributions of this study, to compare the above-mentioned methods are: firstly, to allow readers to understand the principle of analysis based on the difference between perception and expectation, and to help readers to select the most appropriate analysis method based on the available data.

## 2. Materials and Methods

### 2.1. Procedure

Four procedures were used in this study, as shown in Figure 2, namely, questionnaire development, data collection, data analysis and discussion, and conclusion and implementation. For discussion (presented by a two-way arrow), the similarities and differences of clustering based on FA and CA were first discussed. Then, group forming from the quadrants of IPA was discussed. Lastly, the agreements and differences between FA and IPA were discussed. This research was approved by the Ethics Committee for Researches Involving Human Subjects, Suranaree University of Technology (COA.73/2561).

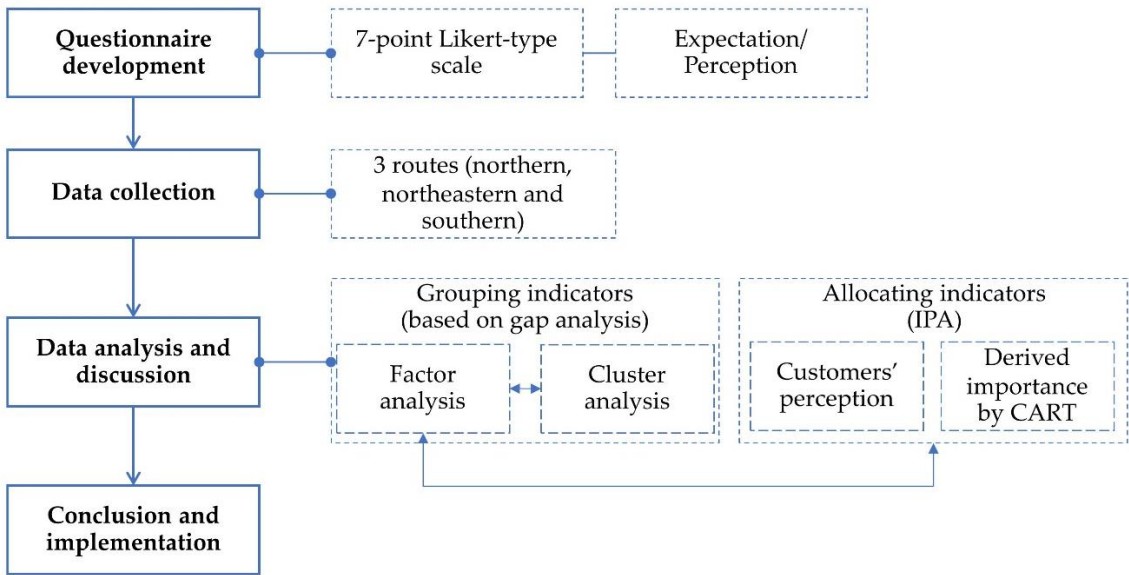

**Figure 2.** The four procedures of this study. CART, classification and regression tree. IPA, importance-performance analysis.

### 2.2. Material and Data Collection

2.2.1. Questionnaire Development

There were two parts to the questionnaire, specifically: (a) the general data and traveling behavior of the respondents, such as gender, age, education, income, occupation, and frequency of using rail services, and (b) the service quality indicators of the intercity rail services, which included 45 indicators separated into 4 factors based on the literature review (i.e., vehicles, staff, services, and station). There were two elements to each indicator, namely, the expectation and the perception. Each indicator was evaluated using a seven-point Likert-type scale [24], where a value of 1 indicated an absolute lack of quality (0%), of 4 a low quality (40%), and of 7 a very high quality (100%).

We verified the content validity of the questionnaire using the index of item-objective congruence (IOC) with five specialists, and we considered questions with an IOC value higher than 0.50. The IOC values of the questions included in the questionnaire were between 0.54 and 1.00.

### 2.2.2. Participants and Data Collection

Hair et al. [25] recommended, in terms of sample size for model development, "Minimum sample size—500: Model with large numbers of constructs, some with lower communalities, and/or having fewer than three measured items." The total sample size in this research was 615, and the respondents were intercity rail passengers of three intercity rail routes in Thailand, including the northern route (Hua Lamphong-Chiang Mai), the northeastern route (Hua Lamphong-Ubon Ratchathani), and the southern route (Hua Lamphong-Su-ngai Kolok).

There are three major rail routes in Thailand, all of which were considered: the northern line, the northeastern line (separated into two sub-major lines, the Nongkhai sub-major line and the Ubon Ratchatani sub-major line), and the southern line. The design of data surveying was dependent upon on the geography of Thailand and its intercity train routes. There are three major regions in Thailand, namely, the north region, the south region, and the northeast region, of which the northeast region has the largest population, and the largest number of passengers from this region traveled to the capital city (Bangkok).

The surveying of service quality was separated based on these three regions, and the population group targeted was the intercity train passengers of said regions. There were 36,895,992 passengers in 2018 (101,085 passengers per day). The survey was done by collecting data at the railway stations in provinces that represented each major line, including Phitsanulok and Chiang Mai on the northern line, Nakhon Ratchasima and Ubon Ratchathani on the northeast line, and Prachuap Khiri Khan and Nakhon Si Thammarat on the southern line. The percentage of respondents was 34.1%, 32.5%, and 33.3% for the north, northeast, and southern line, respectively (from 615 respondents). The statistical data of the respondents are shown in Table 1. There were three train classes included: (1) the special express train (SP EXP), with 39.51% of the total passengers, which is the highest speed train and stops at only the main stations, and which has the highest ticket price in Thailand; (2) the express train (EXP), with 21.63% of the total passengers, which is slower than that of SP EXP because it stops at more stations; and (3) the rapid train (RAP), with the remaining 38.86% of passengers, which stops at main stations as well as sub-stations, and thus has the lowest ticket price and the longest traveling time. For an overview of the respondents, 64.88% were female, which makes sense as the proportion of females is higher than that of males in Thailand. In terms of income, most of the respondents (46.99%) earned less than 10,000 THB per month. The price of the intercity train is lower than other public transportation and personal cars; therefore, trains are the most popular mode of travel among low-income passengers. This is in accordance with the occupations of the respondents, as the study showed that 36.1% were students. The top three traveling objectives were visiting their hometown, tourism, and visiting family at 37.24%, 28.29%, and 19.84%, respectively. In terms of the frequency of traveling, most of the respondents stated that they travel once a year (29.27%), with visiting hometowns as their major traveling objective. This was followed by a travel frequency of two to three times a month (19.35%). The objective and frequency of traveling are distributed according to the actual situation; thus, it can be concluded that the description of respondents can be utilized for a service quality questionnaire.

### 2.3. Analysis Methods

#### 2.3.1. Factor Analysis

Factor analysis was applied with two objectives, as follows:

(1) Exploratory factor analysis (EFA) was used to form indicator groups by focusing on a smaller number of factors than number of indicators. The factors (i.e., latent variables) were groups of indicators (i.e., observed variables). Examples of the factors in this research are vehicles, staff,

services, and stations, which were grouped by factor analysis (FA). EFA is an analysis method used to survey and define common factors in order to explain correlations among observed variables. In other words, the results from EFA can reduce the observed variables by creating new variables in the form of common factors. Researchers usually utilize this method if there is not any clear supporting theory in terms of a correlation between measurement components and the score from each measurement indicator [7].

(2) Confirmatory factor analysis (CFA) was used to verify the loading of each indicator. CFA was utilized when the researchers knew that the indicators were components of factors based on theory or the literature review [26]. However, there has never been a study of the composition of service quality indicators of the trains in Thailand; therefore, CFA was utilized based on the correlation structure from the results of EFA. Most research works in the transportation field focus on CFA (e.g., Jomnonkwao et al. [7], Watthanaklang et al. [17], and Ratanavaraha et al. [26]). This study applied both EFA and CFA to reduce the number of qualitative indicators of the Thai intercity rail services. Indicator group forming was considered, and indicator loading, obtained from the analyses, was utilized to create appropriate guidance to improve intercity rail services.

### 2.3.2. Cluster Analysis

While most studies analyze clustering by respondents, this study analyzed, for comparison with clustering, using the factor analysis method. Thus, we conducted a cluster analysis of group-forming variables. Firstly, the appropriate number of groups was defined by K-means clustering. Then, the hierarchical clustering technique was utilized to form service indicator groups by the Ward method [20,27]. The process of the Ward method is as follows: calculate the mean of every variable in the cluster, and then calculate the square Euclidean distance value from the cluster mean of each member; conclude these distances in every member, and sum the squared values of the cluster distances from the two combined groups to be minimized. The results of the hierarchical clustering technique can be presented in a dendrogram that easily allows which ports belong to each group to be seen [28]. The horizontal axis of the dendrogram represents the distance or dissimilarity between clusters, while the vertical axis represents the objects and clusters. The horizontal position of the split, shown by the short vertical bar, gives the distance (dissimilarity) between the two clusters. We focused on similarity and clustering, and thus we started from the clustering definition of the K-means results [20]. According to a previous case study of intercity rail in China, Yao et al. [28] clustered train stations in each city based on origin–destination to cluster the structure of the ticket price.

### 2.3.3. Importance-Performance Analysis (IPA)

The importance–performance analysis (IPA) technique was developed by Martilla and James [21], and is used for service quality evaluation, which is usually applied in the form of a questionnaire regarding expectations and satisfaction. In this study, the data were collected from service users via a questionnaire. The important characteristics of IPA are to cluster data from two dimensions, namely, customer satisfaction or performance (X-axis) and importance (Y-axis), which are measured from self-states (rating scales, etc.) or derived importance (multiple regression weight) [29]. In the present study, these were separated into four quadrants (Figure 3) as follows:

1. "possible overkill" (low importance/high satisfaction): enterprises can consider reducing development in these perspectives;
2. "keep up the good work" (high importance/high satisfaction): enterprises can consider continuing with these strategies;
3. "low priority" (low importance/low satisfaction): the criteria in this quadrant have a low focus, and improvement is not needed;
4. "concentrate here" (high importance/low satisfaction): the enterprise should urgently improve the criteria in this quadrant [30].

In previous research works, IPA was applied to find the correlation between importance and satisfaction, for example, evaluation of the Marine-Park hinterland of western Australian [31], tourism evaluation [32–34], evaluation of the service received by patients [35,36], and evaluation of the service quality of public transportation in Brazil [30,37].

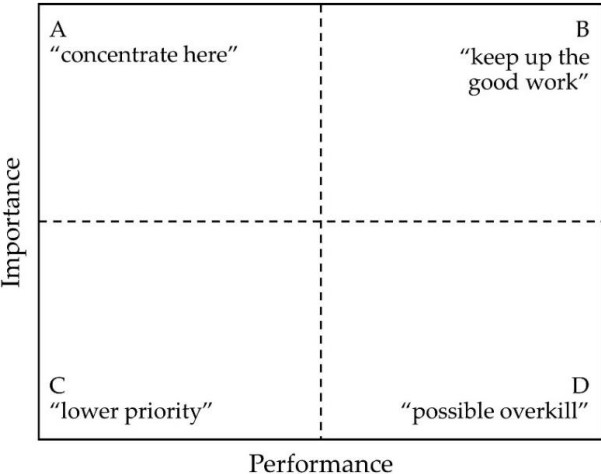

**Figure 3.** Importance–performance analysis (IPA).

The method was applied to analyze the data, which were separated into two processes as follows: (i) Variable clustering by FA and CA. These two methods aggregated question items to be gaps. The gap was calculated from $gap_i = perception_i − expectation_i$, where $i$ is the quality indicator of the intercity rail service [38–40]. The meaning of each indicator is shown in Table 2, and the descriptive statistics are shown in Table A1. The average gap in each question item is shown in Figure 4; positive gap values show that the perception was higher than the expectation of the indicators. The highest positive gap values were G32 and G10, which were indicators for "suitable ticket prices" and "enough train carriages of appropriate size", respectively. The highest negative gap value was G43, "other convenient infrastructure, e.g., WiFi". (ii) The position of each indicator was defined into one of the quadrants; IPA was applied in this process.

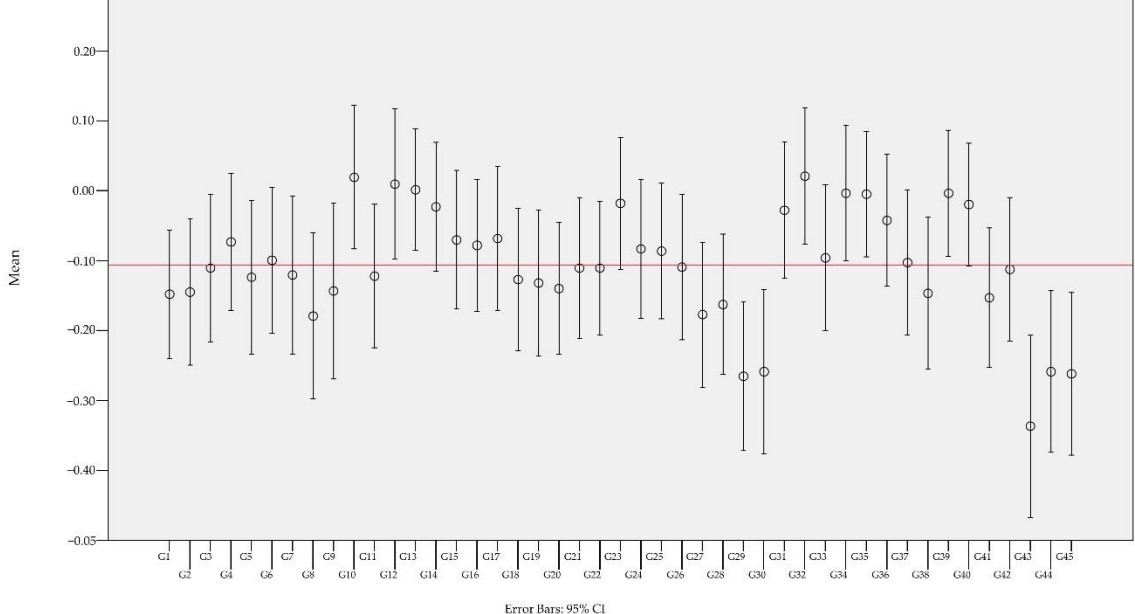

**Figure 4.** Average of gap. Note: red line was determined for overall gap of rail service.

**Table 1.** Respondents' demographics.

| Characteristics | | SP EXP | | EXP | | RAP | | Total | % |
|---|---|---|---|---|---|---|---|---|---|
| | | Freq. | % | Freq. | % | Freq. | % | | |
| Region | Northern line | 40 | 6.50% | 35 | 5.69% | 135 | 21.95% | 210 | 34.15% |
| | Northeastern line | 134 | 21.79% | 49 | 7.97% | 17 | 2.76% | 200 | 32.52% |
| | Southern line | 69 | 11.22% | 49 | 7.97% | 87 | 14.15% | 205 | 33.33% |
| Gender | Male | 75 | 12.20% | 52 | 8.46% | 89 | 14.47% | 216 | 35.12% |
| | Female | 168 | 27.32% | 81 | 13.17% | 150 | 24.39% | 399 | 64.88% |
| Education | Elementary school | 5 | 0.81% | 2 | 0.33% | 21 | 3.41% | 28 | 4.55% |
| | Primary | 5 | 0.81% | 6 | 0.98% | 27 | 4.39% | 38 | 6.18% |
| | High school | 66 | 10.73% | 30 | 4.88% | 68 | 11.06% | 164 | 26.67% |
| | High vocational | 81 | 13.17% | 56 | 9.11% | 66 | 10.73% | 203 | 33.01% |
| | Bachelor | 78 | 12.68% | 36 | 5.85% | 56 | 9.11% | 170 | 27.64% |
| | Master | 7 | 1.14% | 3 | 0.49% | 1 | 0.16% | 11 | 1.79% |
| | Doctoral | 1 | 0.16% | 0 | 0.00% | 0 | 0.00% | 1 | 0.16% |
| Salary (THB/Month) | <10,000 | 127 | 20.65% | 44 | 7.15% | 118 | 19.19% | 289 | 46.99% |
| | 10,000–14,999 | 39 | 6.34% | 28 | 4.55% | 53 | 8.62% | 120 | 19.51% |
| | 15,000–19,999 | 22 | 3.58% | 31 | 5.04% | 37 | 6.02% | 90 | 14.63% |
| | 20,000–24,999 | 17 | 2.76% | 11 | 1.79% | 18 | 2.93% | 46 | 7.48% |
| | 25,000–29,999 | 25 | 4.07% | 12 | 1.95% | 10 | 1.63% | 47 | 7.64% |
| | >30,000 | 13 | 2.11% | 7 | 1.14% | 3 | 0.49% | 23 | 3.74% |
| Occupation | Government/state enterprises | 54 | 8.78% | 36 | 5.85% | 22 | 3.58% | 112 | 18.21% |
| | Company employees | 29 | 4.72% | 31 | 5.04% | 46 | 7.48% | 106 | 17.24% |
| | Personal business | 18 | 2.93% | 12 | 1.95% | 29 | 4.72% | 59 | 9.59% |
| | Farmers | 4 | 0.65% | 3 | 0.49% | 4 | 0.65% | 11 | 1.79% |
| | Students | 117 | 19.02% | 30 | 4.88% | 75 | 12.20% | 222 | 36.10% |
| | Other | 21 | 3.41% | 21 | 3.41% | 63 | 10.24% | 105 | 17.07% |
| Purpose | Hometown | 99 | 16.10% | 36 | 5.85% | 94 | 15.28% | 229 | 37.24% |
| | Traveling | 64 | 10.41% | 43 | 6.99% | 67 | 10.89% | 174 | 28.29% |
| | Working | 29 | 4.72% | 15 | 2.44% | 14 | 2.28% | 58 | 9.43% |
| | Visiting relations | 43 | 6.99% | 35 | 5.69% | 44 | 7.15% | 122 | 19.84% |
| | Other | 8 | 1.30% | 4 | 0.65% | 20 | 3.25% | 32 | 5.20% |
| Frequency | once a week | 40 | 6.50% | 14 | 2.28% | 51 | 8.29% | 105 | 17.07% |
| | once every two-weeks | 20 | 3.25% | 8 | 1.30% | 28 | 4.55% | 56 | 9.11% |
| | once a month | 23 | 3.74% | 9 | 1.46% | 29 | 4.72% | 61 | 9.92% |
| | once every two months | 49 | 7.97% | 22 | 3.58% | 48 | 7.80% | 119 | 19.35% |
| | once every 4–6 months | 31 | 5.04% | 25 | 4.07% | 34 | 5.53% | 90 | 14.63% |
| | once a year | 80 | 13.01% | 55 | 8.94% | 49 | 7.97% | 184 | 29.92% |
| Total | | 243 | 39.51% | 133 | 21.63% | 239 | 38.86% | 615 | 100.00% |

**Table 2.** Factor analysis results.

| Variable | Description | EFA | | | | CFA | | | | |
|---|---|---|---|---|---|---|---|---|---|---|
| | | Communalities | Loading | Explained Variance (%) | Cronbach's $\alpha$ | Loading | *t*-Value | Error Variance | CR | AVE |
| | Factor 1 | | | 15.900 | 0.973 | | | | 0.997 | 0.743 |
| G1 | Decent vehicle body appearance. | 0.611 | 0.703 | | | 0.715 | 33.843 | 0.021 | | |
| G2 | No engine noise disturbance when inside the train. | 0.657 | 0.704 | | | 0.773 | 43.899 | 0.018 | | |
| G3 | Neat and clean train interior. | 0.652 | 0.715 | | | 0.767 | 42.609 | 0.018 | | |
| G4 | Cool but convenient interior temperature. | 0.529 | 0.654 | | | 0.673 | 28.838 | 0.023 | | |
| G5 | Clean train seats. | 0.634 | 0.704 | | | 0.756 | 40.638 | 0.019 | | |
| G6 | Train seats with an appropriate space between two seats in a row. | 0.579 | 0.654 | | | 0.735 | 36.961 | 0.020 | | |
| G7 | Train seats are adjustable, of a suitable size, and convenient to use. | 0.611 | 0.670 | | | 0.755 | 40.366 | 0.019 | | |
| G8 | Variety of entertainment devices available in good working condition. | 0.651 | 0.631 | | | 0.802 | 50.731 | 0.016 | | |
| G9 | Clean, convenient toilets and washrooms. | 0.626 | 0.610 | | | 0.788 | 47.286 | 0.017 | | |
| G10 | Enough train carriages of appropriate size. | 0.531 | 0.577 | | | 0.715 | 34.013 | 0.021 | | |
| G11 | Properly functioning windows and doors. | 0.576 | 0.609 | | | 0.739 | 37.577 | 0.020 | | |
| G12 | Onboard food and drink services. | 0.543 | 0.528 | | | 0.702 | 32.197 | 0.022 | | |
| | Factor 2 | | | 16.214 | 0.935 | | | | 0.997 | 0.733 |
| G13 | Neat and clear train crews. | 0.515 | 0.631 | | | 0.648 | 25.985 | 0.025 | | |
| G14 | On-time, accurate delivery of advertised services. | 0.634 | 0.703 | | | 0.732 | 36.002 | 0.020 | | |
| G15 | Crew able and willing to solve onboard problems. | 0.685 | 0.734 | | | 0.788 | 46.721 | 0.017 | | |
| G16 | Crews deliver information before the start of every service. | 0.618 | 0.730 | | | 0.732 | 36.065 | 0.020 | | |
| G17 | Crews can deliver fast and accurate services. | 0.671 | 0.730 | | | 0.784 | 45.572 | 0.017 | | |
| G18 | Crews are always willing to help. | 0.598 | 0.664 | | | 0.747 | 38.296 | 0.020 | | |
| G19 | Crews respond willingly to all passenger requests. | 0.636 | 0.692 | | | 0.779 | 44.718 | 0.017 | | |
| G20 | Crew behavior makes passengers confident about the service. | 0.579 | 0.648 | | | 0.752 | 39.458 | 0.019 | | |
| G21 | Crews deliver services politely. | 0.613 | 0.663 | | | 0.776 | 43.942 | 0.018 | | |
| G22 | Crews are knowledgeable and can provide accurate, complete information. | 0.608 | 0.528 | | | 0.723 | 34.699 | 0.021 | | |
| G23 | Attentive personal passenger service. | 0.533 | 0.504 | | | 0.653 | 26.603 | 0.025 | | |
| G24 | Passenger service willingness by crews. | 0.589 | 0.525 | | | 0.683 | 29.577 | 0.023 | | |
| | Factor 3 | | | 12.397 | 0.914 | | | | 0.996 | 0.717 |
| G25 | Passenger service is important to the State Railway of Thailand. | 0.580 | 0.514 | | | 0.673 | 28.375 | 0.024 | | |
| G26 | Crews understand special passenger requirements. | 0.566 | 0.522 | | | 0.674 | 28.545 | 0.024 | | |
| G27 | Train timetables and train frequencies are suitable. | 0.665 | 0.691 | | | 0.742 | 37.466 | 0.020 | | |
| G28 | Safe traveling conditions (without accidents or broken-down trains). | 0.621 | 0.625 | | | 0.763 | 41.228 | 0.019 | | |
| G29 | Security system available to prevent crime. | 0.624 | 0.641 | | | 0.746 | 38.336 | 0.019 | | |
| G30 | Punctuality. | 0.558 | 0.480 | | | 0.731 | 35.858 | 0.020 | | |
| G31 | Enough ticket counters. | 0.514 | 0.469 | | | 0.722 | 34.566 | 0.021 | | |
| G32 | Suitable ticket prices. | 0.501 | 0.518 | | | 0.696 | 31.086 | 0.022 | | |
| G33 | Suitable onboard meal prices. | 0.555 | 0.569 | | | 0.706 | 32.399 | 0.022 | | |
| G34 | Services received as agreed on the ticket. | 0.551 | 0.573 | | | 0.713 | 33.314 | 0.021 | | |

**Table 2.** *Cont.*

| Variable | Description | EFA | | | | CFA | | | | |
|---|---|---|---|---|---|---|---|---|---|---|
| | | Communalities | Loading | Explained Variance (%) | Cronbach's $\alpha$ | Loading | *t*-Value | Error Variance | CR | AVE |
| | Factor 4 | | | 14.854 | 0.926 | | | | 0.996 | 0.727 |
| G35 | Terminal provides enough trip guidance and information. | 0.563 | 0.549 | | | 0.715 | 33.204 | 0.022 | | |
| G36 | Provides information when train timetable changes. | 0.559 | 0.575 | | | 0.718 | 33.851 | 0.021 | | |
| G37 | Provides connecting information to other public transportation. | 0.603 | 0.574 | | | 0.744 | 37.645 | 0.020 | | |
| G38 | Provides a complaint channel. | 0.588 | 0.589 | | | 0.744 | 37.611 | 0.020 | | |
| G39 | Suitable terminal sizes. | 0.567 | 0.673 | | | 0.678 | 28.797 | 0.024 | | |
| G40 | Suitable terminal locations, ease of dis/embarking. | 0.535 | 0.641 | | | 0.665 | 27.493 | 0.024 | | |
| G41 | Cleanliness of terminals. | 0.648 | 0.720 | | | 0.752 | 38.969 | 0.019 | | |
| G42 | Enough seats inside the terminals. | 0.593 | 0.659 | | | 0.743 | 37.436 | 0.020 | | |
| G43 | Other convenient infrastructure, e.g., Wi-Fi. | 0.612 | 0.653 | | | 0.750 | 38.553 | 0.019 | | |
| G44 | Enough car parks at the terminals. | 0.609 | 0.658 | | | 0.741 | 36.899 | 0.020 | | |
| G45 | Security system available to prevent crime at terminals. | 0.620 | 0.668 | | | 0.747 | 38.083 | 0.020 | | |

Note: Goodness of fit for EFA: Kaiser–Meyer–Olkin (KMO) measure of sampling adequacy = 0.974, Bartlett's test approx. $\chi^2$ = 19,219.364, degrees of freedom (df) = 990, *p* < 0.000. Goodness of fit for CFA: Chi-square test of model fit $\chi^2$ = 2319.158, df = 929, comparative fit index (CFI) = 0.926, Tucker–Lewis index (TLI) = 0.921, root mean square error of approximation (RMSEA) = 0.049 (0.047–0.052), standardized root mean square Residual (SRMR) = 0.043. EFA, exploratory factor analysis; CFA, confirmatory factor analysis; CR, composite reliability; AVE, average variance extracted.

## 3. Results

### 3.1. Factor Analysis

Table 2 presents the factor analysis. The analysis results of the EFA were reliable and can be accepted. We obtained a Kaiser–Meyer–Olkin (KMO) measure of 0.974, which was in the "very good" category. Furthermore, the Bartlett's test results had significant reliability, higher than 99%. For the results, the number of groups was separated into four. The correlation values among indicators were at a good level. The Cronbach's α values were in the 0.914–0.973 range and the explained variance percentage was in the 12.397–16.214 range [41]. The clustering results included: (1) the vehicle factor (e.g., no engine noise disturbance when inside the train); (2) the staff and crew factor (e.g., crew able and willing to solve onboard problems); (3) the service factor (e.g., train timetables and train frequencies are suitable); and (4) the infrastructure and station factor (e.g., cleanliness of terminals).

The CFA analysis results were considered from the perspective of goodness of fit. The overall results were at the good level, based on the ratio value of $\chi^2/df < 3$, comparative fit index (CFI), Tucker–Lewis index (TLI) > 0.92, root mean square error of approximation (RMSEA), and standardized root mean square residual (SRMR) < 0.05 [42,43]. The results of secondary CFA verified that all indicators were clustered into four groups. When considering weighted values (in Figure 5), it was found that the loading factor of vehicle was the maximum (0.898), followed by service, staff, and infrastructure/station, with factor loadings at 0.891, 0.859, and 0.845, respectively. For the weighting value of each indicator, the suitability was considered from the composite reliability (CR) value in the range of 0.996–0.997 and an average variance extracted (AVE) value in the range of 0.717–0.743 [17].

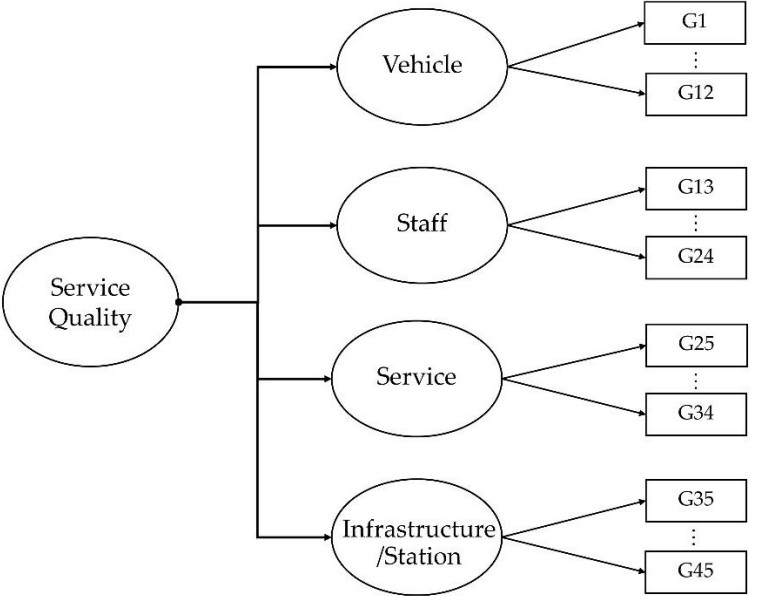

**Figure 5.** Confirmatory factor analysis result.

### 3.2. Cluster Analysis

The results of cluster analysis were analyzed by ward linkage [20]. A dendrogram is shown in Figure 6. The number of clustered groups was operated by K-means; it was found that the most appropriate number of groups was three to five groups. The dendrogram is presented in the pattern of hierarchical cluster analysis; the Y-axis represents the distance among groups, and the X-axis represents the indicators. The k-means results found that the final number of groups was four. Group 1 was compounded from G13 to G26, Group 2 from G1 to G12, Group 3 from G43 to G45, and Group 4 from G27 to G42.

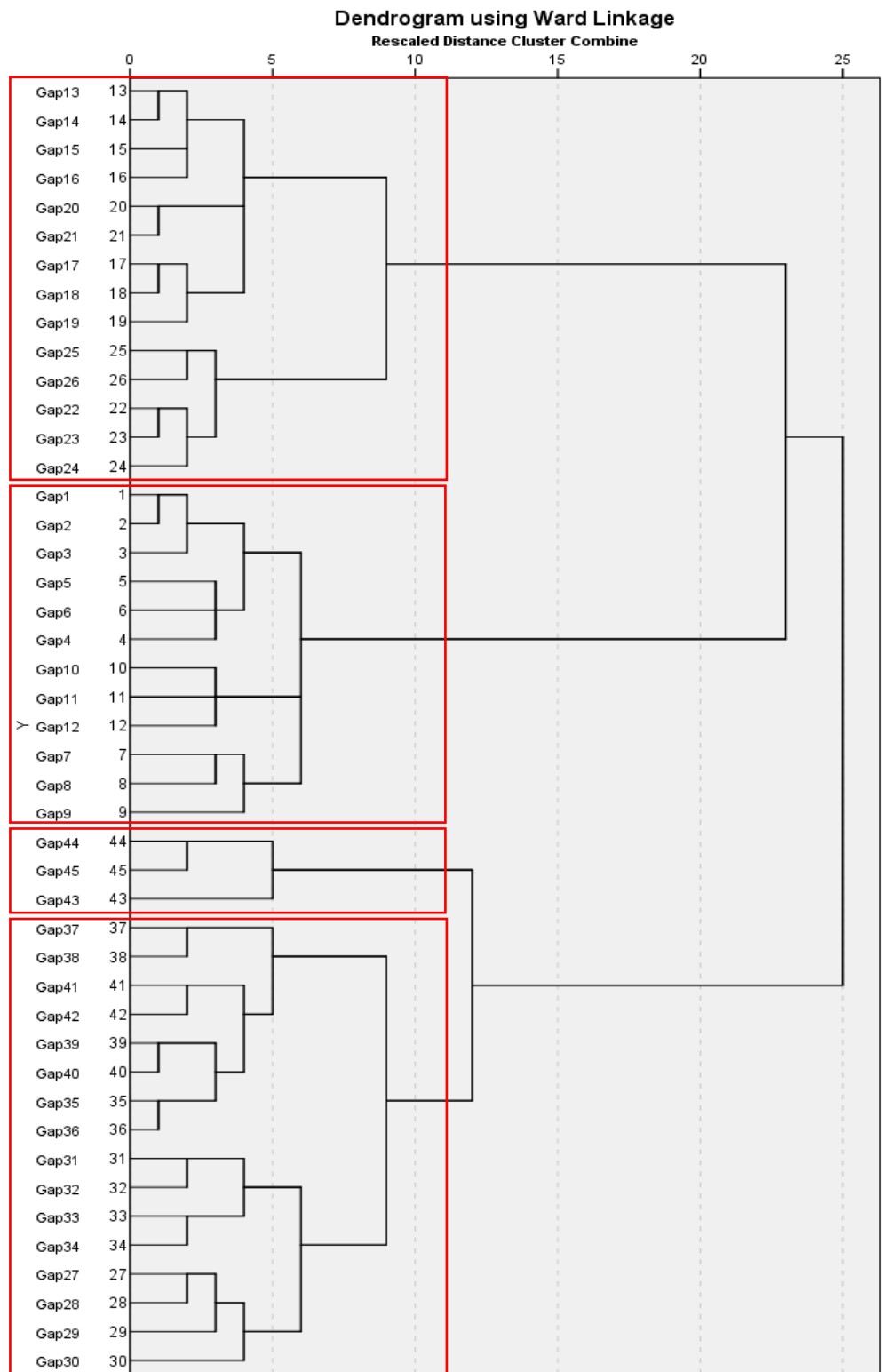

**Figure 6.** Cluster analysis based on the gap variables.

### 3.3. Importance-Performance Analysis (IPA)

The results of the IPA showed four quadrants. The Y-axis intercept was the average of perception from all indicators, which was 4.889. The X-axis intercept was the average of the variable importance level percentage at 29.49%, which was from the CART (classification and regression tree) analysis (Table A2). From the overall perspective, it was found that most of the indicators were in the "keep

up the good work" and "possible overkill" quadrants. For the "keep up the good work" quadrant, the obvious indicators were G14, "cool but convenient interior temperature", and G25, "passenger service is important to the State Railway of Thailand". For the "concentrate here" quadrant, the most important indicator with average satisfaction lower than the total average was G45, "security system available to prevent crime at terminals", and another obvious indicator was G29, "security system available to prevent crime".

Considering the analysis results of three methods, the service quality indicator grouping was analyzed by FA and CA. The results found that there was a difference in both the number of groups and indicators in each group. The grouping of the FA method was based on patterns of variation (correlation) of data, while the CA method created groups with distance (proximity) consideration. The comparison of the grouping methods was considered from the meaning of factors and the previous research. The IPA was utilized to cluster indicators based on the urgency of taking action. The advantages of IPA were easy and uncomplex analysis procedure using direct data from the questionnaire without indicators correlation evaluation as the FA [44].

## 4. Discussion

### 4.1. Grouping Service Quality Indicators

The results of the grouping were analyzed by two methods, namely, cluster analysis (CA) and factor analysis (FA). Considering the results of CA, it was found that the appropriate number of groups was three to five groups, and then the final consideration was four groups. For the results of FA, the indicators were separated into four groups as well. When comparing with the SERVQUAL theory [45], which was applied to evaluate user satisfaction from user expectation and perception with five sub-groups, it was concluded that there was not accordance, because of the different attitudes of samples and the different complete services. However, many research works did not operate group forming similar to the SERVAUL theory for intercity rail satisfaction surveying, e.g., Sivilevičius et al. [12], which included railway track, railway trip planning, safety, and price of trip ticket. The overview of both methods is discussed as follows:

Group 1: The results of the FA were G1–G12, which obviously accorded with CA. When considering the indicators of this group, they were presented in terms of vehicle characteristics, e.g., clean train seats, decent vehicle body appearance, clean toilets, and onboard food and drink services. This group can be verified by the studies of Ratanavaraha et al. [11] and Champahom et al. [41]. They grouped the quality of public transit indicators and found that train car factors constituted the most important group from the perspective of passengers.

Group 2: The results of the FA were G13–G24, while the results of CA were G13–G26. The overview consideration found that these were indicators about services from staff, e.g., neat and clear train crews, crews can deliver fast services, and crews are knowledgeable to provide information. This accorded with the study of Aydin et al. [46], who formed indicator groups to analyze the correlation between satisfaction of rail transit system passengers. The study's results found that the reception of staff had a medium-weighted value and was very high for customer satisfaction relationships. When considering comparison groups between FA and CA, we found that CA had higher validity. Thus, both G25, "passenger service is important to the State Railway of Thailand," and G26, "crews understand special passenger requirements", should be placed in the "staff" group.

Group 3: The results of the FA were G25–G34, while the results of CA were G27–G42. There was an obvious difference in this group. The duplicate indicators of both methods were the seven indicators of G27–G34, which were mostly service indicators, e.g., train frequency, safety, punctuality, suitable ticket counters, suitable ticket prices, suitable onboard meal prices, and services received as agreed on the ticket [47]. The remaining indicators from the results of the CA (G35–G42) were the summation of information indicators (e.g., provides information when train timetable changes, provides a complaint channel) [47] and a few terminal indicators (e.g., suitable terminal sizes, suitable terminal location,

cleanliness of terminals, and enough seats inside the terminals) [15]. It can be concluded that with this group, the results of the FA were more suitable than that of the CA. However, G25 and G26 should be in Group 2.

Group 4: The results of the FA were G35–G45, while the results of the CA were G43–G45. The group of indicators from the FA analysis combined indicators from two groups, namely, the information group (G35–G38; e.g., provides a complaint channel, or provides connecting information to other public transportation) [48], and the station group (G39–G45; e.g., suitable terminal sizes, enough seats inside the terminals, and enough car parks at the terminals) [49]. There were three indicators from the results of the CA, because the information variables were placed in the previous group. These three indicators were not placed in this group, because their perception average was absolutely different, as shown in Figure 4. There were extremely negative values of G43–G45.

### 4.2. Notable Service Quality Indicators

The study results from the FA can be translated into policy by considering loading from the model of confirmatory factor analysis. Importance-performance analysis (IPA) can be interpreted for policy by considering service quality (SQ) indicators placed in important quadrants, such as "keep up the good work" and "concentrate here". In the dimension of quality improvement, we considered accordance with both FA (top 15 highest factor loadings) and IPA. It was found that there were three indicators that needed urgent improvement. The first two indicators were G3, "neat and clean train interior", and G11, "properly functioning windows and doors". The reasons were due to the age of Thai train cars; most doors and windows could not open or close, and the train car interiors were old-looking, which led to passengers being unsatisfied. This accorded with the study of Alpu [48], who stated that cleanliness and the interior were the first to draw a customer's attention. Furthermore, De Oña et al. [50] found that the usability of windows and doors was the first factor of importance. Another indicator was G29, "security system available to prevent crime". This indicator obviously represented security onboard, which accorded with the findings of de Oña et al. [51], who found that security strongly affected passengers during their vacations.

For interpretation of the other dimensions, e.g., the point that transportation should develop efficiency, there was only one IPA result that can be interpreted in this dimension. Consider that Figure 7 shows that many indicators were placed in the "keep up the good work" quadrant, e.g., G14, "on-time, accurate delivery of advertised services", G21, "crews deliver services politely", and G25, "passenger service is important to the State Railway of Thailand". The top three most outstanding indicators in the "keep up the good work" quadrant were in the group of crew and staff. It can be concluded that this strength, which should continue developing, was the service of staff. This was reasonable, because the service attitude and behavior of staff toward passengers highly affected service quality perception. Indeed, many studies found that the dimension of staff highly influenced user satisfaction [22,46].

Another interpretation from FA was the highest loading consideration for creating policy recommendations, which should be urgently developed. The top three highest loadings were in the group of train cars, including G8, G9, and G2, which were the dimensions of entertainment devices, cleanliness of toilets, and engine noise disturbance [48,50]. These three indicators were emphasized by the service operator. Thus, passenger satisfaction would increase if the quality of these dimensions was improved.

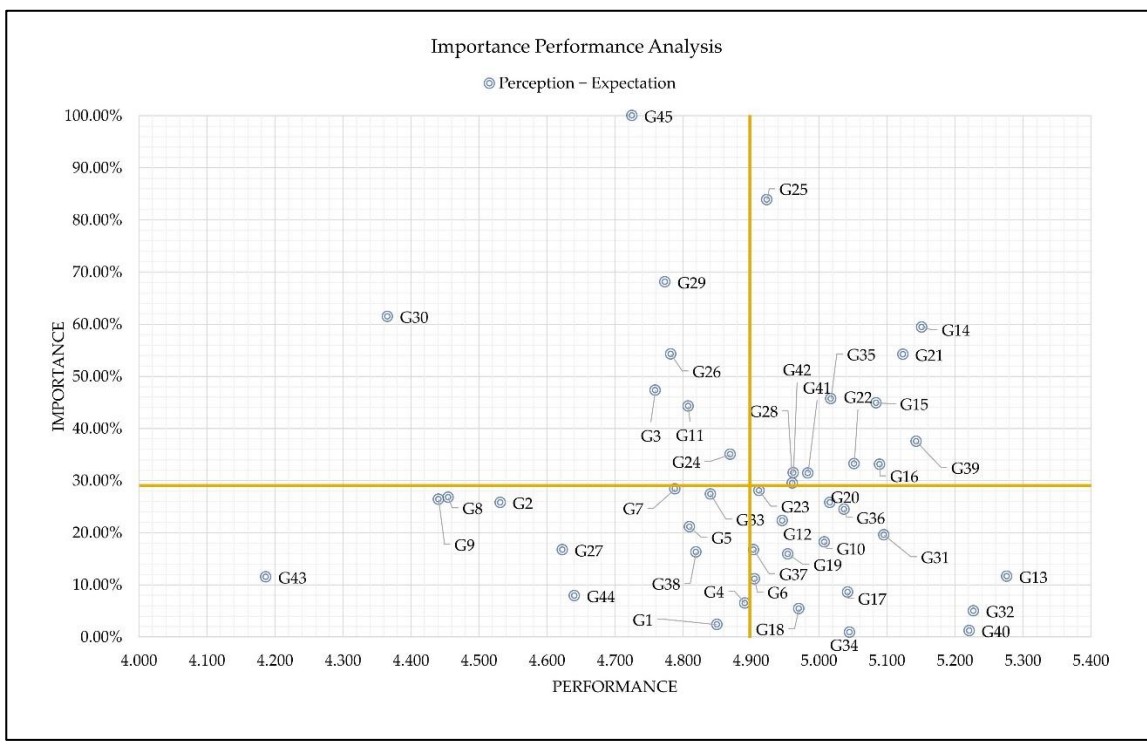

**Figure 7.** Importance-performance analysis (IPA).

## 5. Conclusions and Implementation

The objectives of this study were to apply various analysis methods to discuss the strengths and weaknesses of each method. The discussion led to the creation of a roadmap of increasing rail transportation passenger satisfaction in Thailand. This is a pertinent quality improvement, which could attract people to use intercity rail.

The three analysis methods included factor analysis (FA), cluster analysis (CA), and importance–performance analysis (IPA). The methods that could form variables into groups were FA and CA; the forming resulted in four groups that were equal to each other. There were three groups that had different group suitability—one group of CA and two groups of FA could achieve better group forming. Thus, it can be concluded that FA can achieve better group forming than CA. The advantage of FA is the ability to compare loadings, which is related to the importance of the indicator group. On the other hand, in order to form an intercity rail service indicator group that considers strengths and weaknesses, only the IPA method could achieve this with simple interpretation of the results. However, the analysis results of the FA had the ability to consider the loading factor. Thus, an indicator with a high loading factor can be developed as a policy recommendation.

Applying various methods can result in the identification of very important indicators, which can lead to pertinent and well-defined service efficiency improvement. The IPA analysis results identified indicators that should be urgently attended to in order to improve intercity rail service. Most of them were in the dimension of train car, with the sub-dimensions of neat and clean train interior, properly functioning windows and doors, and a security system. Moreover, the FA analysis results showed that the dimension that should be urgently improved was train car, including engine noise disturbance, entertainment devices, and clean toilets.

The results of the study can be applied as an exploratory study for railway services in Thailand to increase the efficiency and satisfaction of services as follows: (1) The State Railway of Thailand should improve quality by setting up a department that frequently organizes the standard of cleanliness on train cars, which may be done by outsourcing to another company. This is in accordance with the implications of a study regarding the cleanliness perceptions of intercity train passengers in the

Netherlands [14]. Moreover, the study of Lierop et al. [10] confirmed that increasing the cleanliness of rail services can increase passenger satisfaction. (2) The problem of dilapidated windows and doors of train cars is obvious in Thailand because of the age of train cars; thus, the State Railway of Thailand should devise policies on the maintenance of other functions of train cars, and not just in terms of their engines. If windows and doors can be perfectly closed, the noise from outside, which disturbs passengers, can be reduced. In addition to the above-mentioned factors, Sivilevičius et al. [12] stated further that the efficiency of handrails, stairs, doors, and lock facilities are criteria related to the safety of the railway trip. Furthermore, another study found that improvements in the usability of windows and doors were also important for increasing the service quality of rail services [50].

This study aimed to create efficiency recommendations by improving intercity rail service quality indicators, which lacked analysis in terms of the repurchase factor and the word of mouth factor. This study considered intercity rail service in the overall region of the country. The comparison analysis among regions was not in the scope of this study. To consider the differences in services among regions is necessary to test for regional differences by using multi-group analysis. Thus, future works can study these issues.

**Author Contributions:** Conceptualization, S.J.; Data curation, S.J. and T.C.; Formal analysis, T.C.; Funding acquisition, S.J.; Methodology, S.J.; Supervision, V.R.; Writing—original draft, S.J.; Writing—review and editing, V.R. All authors have read and agreed to the published version of the manuscript.

**Funding:** This research and the APC were funded by the Thailand Research Fund (TRF) and Office of the Higher Education Commission, grant number MRG6180052.

**Acknowledgments:** The authors would like to thank the Suranaree University of Technology.

**Conflicts of Interest:** The authors declare no conflict of interest.

## Appendix A

**Table A1.** Descriptive statistics.

| Variables | Min | Max | Mean | Standard Deviation | Skewness | Kurtosis |
|---|---|---|---|---|---|---|
| G1 | −5 | 4 | −0.148 | 1.161 | −0.317 | 2.016 |
| G2 | −5 | 4 | −0.145 | 1.323 | −0.334 | 1.541 |
| G3 | −6 | 5 | −0.111 | 1.335 | −0.229 | 2.152 |
| G4 | −5 | 5 | −0.073 | 1.237 | 0.025 | 1.626 |
| G5 | −6 | 5 | −0.124 | 1.385 | −0.006 | 2.136 |
| G6 | −5 | 5 | −0.099 | 1.317 | −0.126 | 1.907 |
| G7 | −5 | 5 | −0.120 | 1.423 | 0.227 | 1.903 |
| G8 | −6 | 5 | −0.179 | 1.499 | −0.548 | 1.981 |
| G9 | −6 | 5 | −0.143 | 1.589 | −0.615 | 1.852 |
| G10 | −4 | 5 | 0.020 | 1.295 | 0.207 | 1.531 |
| G11 | −5 | 5 | −0.122 | 1.294 | −0.238 | 2.161 |
| G12 | −6 | 5 | 0.010 | 1.360 | 0.029 | 2.097 |
| G13 | −5 | 4 | 0.002 | 1.099 | −0.048 | 1.725 |
| G14 | −6 | 4 | −0.023 | 1.162 | −0.212 | 1.918 |
| G15 | −6 | 4 | −0.070 | 1.254 | −0.335 | 2.431 |
| G16 | −6 | 4 | −0.078 | 1.189 | −0.572 | 2.875 |
| G17 | −6 | 4 | −0.068 | 1.302 | −0.215 | 2.058 |
| G18 | −6 | 6 | −0.127 | 1.289 | −0.037 | 3.760 |
| G19 | −6 | 5 | −0.132 | 1.319 | −0.405 | 2.342 |
| G20 | −6 | 5 | −0.140 | 1.192 | −0.394 | 2.848 |
| G21 | −6 | 4 | −0.111 | 1.273 | −0.391 | 2.547 |
| G22 | −5 | 5 | −0.111 | 1.203 | 0.067 | 2.733 |
| G23 | −6 | 5 | −0.018 | 1.192 | −0.399 | 3.247 |
| G24 | −6 | 4 | −0.083 | 1.255 | −0.816 | 3.723 |
| G25 | −6 | 4 | −0.086 | 1.227 | −0.387 | 2.430 |
| G26 | −5 | 4 | −0.109 | 1.314 | −0.235 | 1.635 |
| G27 | −6 | 5 | −0.177 | 1.310 | −0.939 | 3.402 |
| G28 | −6 | 5 | −0.163 | 1.268 | −0.595 | 2.890 |
| G29 | −6 | 5 | −0.265 | 1.337 | −0.499 | 2.330 |

**Table A1.** *Cont.*

| Variables | Min | Max | Mean | Standard Deviation | Skewness | Kurtosis |
|---|---|---|---|---|---|---|
| G30 | −6 | 5 | −0.259 | 1.487 | −0.916 | 3.200 |
| G31 | −6 | 6 | −0.028 | 1.234 | 0.376 | 3.473 |
| G32 | −4 | 6 | 0.021 | 1.229 | 0.409 | 2.935 |
| G33 | −6 | 5 | −0.096 | 1.318 | −0.238 | 2.451 |
| G34 | −6 | 6 | −0.003 | 1.221 | −0.182 | 4.001 |
| G35 | −6 | 4 | −0.005 | 1.129 | −0.263 | 2.384 |
| G36 | −6 | 4 | −0.042 | 1.192 | −0.566 | 3.428 |
| G37 | −6 | 5 | −0.102 | 1.306 | −0.281 | 3.073 |
| G38 | −6 | 5 | −0.146 | 1.373 | −0.247 | 2.084 |
| G39 | −6 | 4 | −0.003 | 1.139 | −0.326 | 2.575 |
| G40 | −4 | 4 | −0.020 | 1.108 | 0.118 | 1.259 |
| G41 | −6 | 5 | −0.153 | 1.259 | −0.289 | 2.377 |
| G42 | −6 | 5 | −0.112 | 1.297 | −0.199 | 1.956 |
| G43 | −6 | 6 | −0.337 | 1.654 | −0.546 | 1.698 |
| G44 | −6 | 5 | −0.259 | 1.457 | −0.647 | 2.713 |
| G45 | −6 | 5 | −0.262 | 1.467 | −0.423 | 1.671 |

**Table A2.** Variable importance.

| Variables | Mean | Normalized Importance |
|---|---|---|
| G1 | 4.850 | 2.40% |
| G2 | 4.532 | 25.80% |
| G3 | 4.759 | 47.30% |
| G4 | 4.891 | 6.50% |
| G5 | 4.810 | 21.10% |
| G6 | 4.906 | 11.10% |
| G7 | 4.789 | 28.40% |
| G8 | 4.455 | 26.80% |
| G9 | 4.441 | 26.40% |
| G10 | 5.008 | 18.20% |
| G11 | 4.808 | 44.30% |
| G12 | 4.946 | 22.30% |
| G13 | 5.276 | 11.60% |
| G14 | 5.151 | 59.40% |
| G15 | 5.085 | 44.90% |
| G16 | 5.089 | 33.10% |
| G17 | 5.042 | 8.60% |
| G18 | 4.971 | 5.40% |
| G19 | 4.954 | 15.90% |
| G20 | 5.016 | 25.80% |
| G21 | 5.124 | 54.20% |
| G22 | 5.052 | 33.20% |
| G23 | 4.912 | 28.10% |
| G24 | 4.870 | 35.00% |
| G25 | 4.924 | 83.90% |
| G26 | 4.782 | 54.30% |
| G27 | 4.623 | 16.70% |
| G28 | 4.961 | 29.50% |
| G29 | 4.774 | 68.10% |
| G30 | 4.366 | 61.50% |
| G31 | 5.096 | 19.60% |
| G32 | 5.228 | 5.00% |
| G33 | 4.841 | 27.40% |
| G34 | 5.046 | 0.90% |
| G35 | 5.018 | 45.70% |
| G36 | 5.037 | 24.50% |
| G37 | 4.904 | 16.70% |
| G38 | 4.820 | 16.30% |
| G39 | 5.143 | 37.50% |
| G40 | 5.221 | 1.20% |
| G41 | 4.984 | 31.40% |
| G42 | 4.963 | 31.50% |
| G43 | 4.187 | 11.50% |
| G44 | 4.641 | 7.90% |
| G45 | 4.725 | 100.00% |

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
