# Peer review of "Methodologies for Determining the Service Quality of the Intercity Rail Service Based on Users’ Perceptions and Expectations in Thailand"

_sustainability, doi:10.3390/su12104259_

Round 1
Reviewer 1 Report
- General comments:
The paper “Methodologies for Determining the Service Quality of the Intercity Rail Service Based on Users’ Perceptions and Expectations: Application of Cluster Analysis, Factor Analysis, and Importance–Performance Analysis“is a well written article that covers theoretical background and research process to investigate the service quality of the intercity rail service. Overall, the paper explains the combination of statistical methods and its process well, and offers a detail explanation on the statistical results. However, there are three weaknesses in the current paper.
First, the respondence statistics shows that the main respondents of the survey are “female students earning low salary”. It seems that it could be the reasonable reason for G32 and G10 which got the highest positive gap value, and G43 which got the highest negative gap value. In other words, the target respondents of the survey need to be extracted in order to provide more comprehensive results on the service quality.
Second, the survey was conducted in the train station but it doesn’t care how many times the respondents are using the rail service in a month or a year. This characteristic of customers would be so important to provide more reasonable results. Another issue on the classification of survey respondents is relating to the regional difference. Would it be same if the research would be conducted separately by north and south region?
Third, there is no mention about the class of train. The survey results should be different if the respondents could be separated by the train class that they are normally using.
Finally, there are some editorial errors that should be edited from the paper.
- Specific comments:
- Line 45: The color of lines in the Figure 1 should be changed to separate the rail network from the highway network
- Line 178: It should be better if there would be “a unit” in the salary factor.
- Line 180 (Table 2): Conbach’s alpha -> Cronbach’s alpha
Reviewer 2 Report
"I am suggesting you to ask a professional editor to revise your writing. For instance, the following sentences that you used in the abstract are not appropriate: “ The results of FA were clear,” These methods were widely used”…
Moreover, the terms that are used in the method section are not clear. For instance, you have to make it clear what is the difference between factors and indicators.
About the structure of the paper, I feel that you did not do a good job; you had a methodology section followed by a procedure section. In the procedure, you are referring to the conclusion, which is the last part of a paper.
You have to provide a background on the k-means and the dendrogram as well.
In general, I feel that the writing of this paper makes it unqualified for publication. "
Reviewer 3 Report
First, this study is performed exclusively for Thailand. So, that should be mentioned in the title of the paper. The results obtained cannot be applied in another geographical region.
Second, the major weakness is the contribution of the paper in the current body of knowledge. Why do we need to perform this study? What is the gap in the current literature? These questions should be answered clearly at the beginning of any paper. This part is not available in the paper.
Third, too many tables and figures are presented. Some of the tables can be included in the Appendix. In addition, it is required to provide implications of the findings in case of an exploratory study. The authors should provide more details on how the findings can be applied in Thailand and how the reported results are comparable to other existing studies.
Fourth, all of the figures have a very low resolution, which makes it hard to visualize what is presented.
Reviewer 4 Report
The Title of the paper is to long, to many words. In the chapters Methods and Procedures is missing the total number of population who is include in research.
Round 2
Reviewer 1 Report
I've checked all the response to the reviewer's comments but I'm still hesitating to provide the better confirmation. For example, the authors mentioned that there is a clear difference between the North and South region but I couldn't find the results of the study on how they would be different in terms of the service quality of rail service., and/or what methods could be better to measure the service quality in each area.
Reviewer 2 Report
The authors uploaded a PDF version of the manuscript that has many comments in balloons, which makes it difficult to read. If the journal decides to ask them to revise, it would be better if you please ask them to upload a clear PDF and another document that shows how they answered to the comments. Thanks.
What is the difference between two of your sections called Gap in the current literature and the objective of the study? Many of the sentences are in common between these two sections.
In the methodology, you said that the first approach is to do a literature review, which is not acceptable. Even in this part, which locates in Section three, you are saying that the result is provided in section 2.
Again, in the methodology section, you are talking about the literature review. Reviewing the methods and how they were used should not be listed here. As I said before, your structure should be changed to be suitable for a journal publication.
In terms of results, you found different types of factors to be fitted in special groups using the methods. Then, it is not apparent how you compared them. As the way you wrote it, it seems that you incorporated your personal feeling, which is not acceptable.
Lastly, I still feel that the structure of this paper is not suitable, while I see how much effort you put in making it better since the last submission.
Reviewer 3 Report
The authors should provide the revised manuscript without track changes--it is very hard to read what is in the system right now.
Round 3
Reviewer 1 Report
Thanks for the updates. It looks better than previous versions.
Reviewer 2 Report
The comments are addressed.
Reviewer 3 Report
Do another round of proof-reading.